# The First Attempt to Apply an Online Mindfulness Program to Nursing Staff in a Traditional Korean Medicine Clinic in COVID-19 Era: A Case Series

**DOI:** 10.3390/healthcare11010145

**Published:** 2023-01-03

**Authors:** Chan-Young Kwon, Do Hyeon Park

**Affiliations:** 1Department of Oriental Neuropsychiatry, Dong-Eui University College of Korean Medicine, Busan 47227, Republic of Korea; 2Department of Arts Psychotherapy, Myongji University, Seoul 03674, Republic of Korea

**Keywords:** mindfulness, nursing staff, traditional Korean medicine, emotional labor, burnout

## Abstract

The importance of medical personnel’s mental health is emphasized in the COVID-19 era. The characteristics of traditional Korean medicine (KM) may affect the mental health of nursing staff working at KM clinics. In this case series, we report the first attempt to apply an online mindfulness program to the nursing staff in a KM clinic in Korea. For three female nursing assistants, an online mindfulness program consisting of five sessions was offered for two months. After the program, a decrease in emotional labor was observed in two participants, and a decrease in the level of burnout was observed in all participants. One participant showed an increase in their emotional labor level, which was associated with an increase in deep acting. The participants expressed a high level of satisfaction with this program in terms of recommendations for peers and willingness to participate again. As this report is a case series, larger studies are needed to fully evaluate the benefits of the program on emotional labor and burnout of KM clinic nursing staff. However, the potential benefits of emotional labor and burnout, high satisfaction, and some challenges identified in this case series can be considered in future extensions and modifications of the program.

## 1. Introduction

In the coronavirus disease of 2019 (COVID-19) era, the mental health issues of medical personnel became more important. Nursing staff constitutes one of the most at-risk medical personnel in relation to mental health [1,2,3]. Nurses are front-line healthcare workers against COVID-19, and the nature of their jobs makes them more likely to be exposed to mental health threats, such as compassion fatigue, burnout, and secondary trauma [3]. A recent retrospective study analyzing suicide incidence reported to the National Violent Death Reporting System found that nurses, but not physicians, had a significantly higher risk of suicide compared to the general population (relative risk in 2017–2018 of 1.18; 95% confidence intervals of 1.03 to 1.36) [4]. Furthermore, we previously found clinical evidence that nurses working in Korea, where medical services are overcrowded, are exposed to elevated levels of emotional labor [5].

There is a dualized medical system in Korea, comprised of conventional Western medicine (WM) and traditional Korean medicine (KM) [6], and the unique characteristics of KM may affect the mental health of the nursing staff working at KM clinics. According to our review, for example, KM clinic nursing staff had significantly lower job satisfaction compared to WM clinic nursing staff in Korea [7], which could potentially adversely affect their mental health. It is important for nurses’ mental health improvement to be individualized according to the work environment [2,5,7]. In this context, KM clinic staff may be familiar with complementary and integrative medicine, including mind–body modalities such as meditation, mindfulness training, and yoga. There is also clinical evidence that these mind–body modalities may improve some aspects of mental health, including burnout, in nurses [8].

Mindfulness is defined as “the awareness that emerges through paying attention on purpose, in the present moment, and nonjudgmentally to the unfolding of experience moment by moment” [9], by Kabat-Zin, who developed a mindfulness-based stress reduction (MBSR) program. In some systematic reviews, MBSR was reported to be effective in improving mindfulness and self-compassion and reducing anxiety, depression, and stress in healthcare professionals [10] as well as employees [11]. Although closely related to mindfulness-based meditation and often included in the modified MBSR program [12], there is loving-kindness meditation and compassion meditation as distinct meditation practices [13]. These meditations are exercises to strengthen the emotional state of kindness and compassion, and their clinical benefits are supported by neuroendocrine studies, neuroimaging studies, and preliminary clinical studies [13]. Given the work specificity of healthcare professionals, it has been suggested that both mindfulness and loving-kindness meditation can help reduce their compassion fatigue and burnout, increase their well-being, and strengthen their relationships with clients [14,15].

However, there have been no documented attempts including mind–body modalities (e.g., mindfulness, and loving-kindness) to improve the mental health of nursing staff in KM clinic settings. As part of the university’s small business support program, we conducted an online mindfulness program of five sessions for two months, and three individuals working in a KM clinic participated. In this case series, we found beneficial effects and high satisfaction with the program, as well as challenges that need improvement for the nurses who participated in it.

## 2. Materials and Methods

This case series complied with the 2013 CARE checklist [16]. All subjects voluntarily participated in the online mindfulness program with an instructor present. The participants were informed that the results of the pre-post evaluation would be used for research purposes, and participation in this evaluation was not mandatory for participation in the program. The participants were informed that they could withdraw from the program or withdraw from providing data for the study at any time they wished. The participants consented to the use of personal information and participated in the evaluation, and data were collected online and anonymized. Fully informed consent from participants was obtained twice, through an online survey platform, Google Forms (Google, Mountain View, CA, USA), before the program and in writing after the program.

The program involved three nurses from a KM outpatient clinic. The participants were not exposed to shift work. Their main tasks were to respond to patients and caregivers, assist with some KM procedures, including acupuncture, cupping, and moxibustion, dispose of medical waste, and manage patient appointment schedules. All three participants were female nursing assistants; two were in their 30s and one was in their 40s. They had no experience participating in specialized training or practicing meditation and mindfulness. Their clinical experiences were three months, one year and one month, and three years and three months. They had no underlying diseases regarding both physical and psychiatric disorders, and their subjective health status was normal before and after the program. In addition to the three nurses, five undergraduate students from a KM university who were interested in this program participated as observers, but they did not participate in the evaluation.

The program was modified and used as follows based on mindfulness training (Table 1). The purpose of this program was to improve the understanding of the stress response and origin of distress and improve work-related stress by promoting mindfulness training in the daily life of participants. The program was created by certified professionals with PhDs in psychology who have been running these programs for working professionals for many years. In general, this program originates from MBSR but there are some differences in its content and implementation. Specifically, the program was modified based on the analysis of mental health factors in Korean nursing staff previously investigated by our team [5,7,17] and the practitioner’s experience. In the program, compassion and loving-kindness were emphasized, and 3 out of 5 sessions (60%) included the topic. Meanwhile, among the formal practices of MBSR, walking meditation was excluded from this program as it was considered inappropriate due to the characteristics of the online non-face-to-face program. Each session was conducted bi-weekly for a total of five sessions, which were held for two hours from 8–10 p.m., accommodating the nurses’ working schedule at the local KM clinic. It is less frequent and shorter in duration than the MBSR, which typically consists of eight 2.5 h classes per week and one 8 h all-day class. The program was implemented online using Zoom meetings (Zoom Video Communications, San Jose, CA, USA). All participants attended every session of this program, except one participant who could not attend Session 3 for personal reasons (Figure 1).

In Session 1, the concept of mindfulness was explained. Psychological education was also conducted on the definition and purpose of mindfulness and the desired attitude when practicing mindfulness. In addition, participants practiced eating meditation and body scanning. In Session 2, participants practiced sitting meditation (static practice) and mindfulness yoga (dynamic practice). Mindfulness yoga is a kind of meditative movement that emphasizes awareness. Sitting meditation is a typical meditation practice that raises awareness of the five senses, emotions, and thoughts non-judgmentally. Both sitting meditation and yoga exercises are considered the formal practices of MBSR. In Session 3, the concept of compassion was introduced and practiced. Loving-kindness meditation was also practiced for cultivating compassion. In Session 4, psychological education and meditation practice on burnout were introduced under the theme of self-care. Compassion fatigue, commonly experienced by medical workers, such as nurses, was explained, and the stages of compassion fatigue and prevention methods were introduced. In Session 5, how to apply mindfulness practice in daily life was discussed. Each participant established a daily meditation practice plan suitable for them based on their own meditation practice experience and shared it with other participants.

Participants in this program completed the following as a pre-and-post evaluation, including the emotional labor assessment tool described by Lee [18], Ham’s Korean-language version of the Copenhagen Burnout Inventory [19], and the occurrence of medical errors in the past six months (pre-evaluation) or past two months (post-evaluation). These tools have been previously used by the research team to evaluate the mental health of nursing staff working at a university hospital setting in Korea [17]. Furthermore, the post-evaluation included evaluating the willingness to re-enter the program and the recommendation of this program to other nurses on a 1–5 Likert scale. Evaluation results were collected twice (1 day before Session 1 and 1 day after Session 5) using Google Forms and extracted to a Microsoft Excel form (Microsoft, Redmond, WA, USA). Participants’ data were visualized using Microsoft Excel software.

## 3. Case Presentations

According to the evaluation results after participating in the program, two participants, participant 1 (P1) and participant 3 (P3), showed a decrease in the level of emotional labor (P1: from 2.02 to 1.33, P2: from 3 to 2.79), while the other participant, participant 2 (P2), showed an increase (from 3.05 to 3.33). Specifically, P1 and P2 showed a decrease in both aspects of measured emotional labor, employee-focused emotional labor (P1: from 2.17 to 1.33, P3: from 3 to 2.5) and job-focused emotional labor (P1: from 2.25 to 1.25, P3: from 3.25 to 3.13). However, P2 showed an increase in employee-focused emotional labor (from 3.33 to 3.83) and both subscales of employee-focused emotional labor, surface acting (from 3.33 to 3.67) and deep acting (from 3.33 to 4). P2 also showed no change in all three subscales of job-focused emotional labor, including frequency of interactions, duration of interactions, and variety of expressions. P1 and P2 showed an overall decrease across the subscales of employee-focused emotional labor and job-focused emotional labor (Figure 2a,c).

All three participants showed a decrease in the measured burnout levels (P1: From 1.79 to 1.32; P2: From 2.47 to 2.42; P3: From 3.05 to 2.26). In the subscales of burnout, the participants showed an overall decrease, but P2 showed an increase in the subscale of personal burnout (from 3 to 3.14) (Figure 2d,e). Two out of three participants (P1 and P2) experienced medical errors in the past six months where the error was a problem with the acupuncture procedure (missing needle removal). Meanwhile, all three participants answered that they did not experience any medical errors during the two months of the program.

Participants were asked non-mandatory, open-ended questions. One question was, “What has helped you the most while participating in this program?” P1 answered, “Everything was good.” P2 and P3 mentioned specific components of the program. P2 mentioned that the body scanning and loving-kindness meditation were helpful, and P3 mentioned that mindfulness and self-care were helpful. The participants were also asked, “What was the most difficult part of participating in this program?” P1 responded that it was difficult to concentrate on participating in the program because the program was conducted late (i.e., from 8–10 p.m.). P2 responded that it was difficult to participate in yoga due to the lack of body flexibility. P3 responded that it was difficult to concentrate during the meditation process. The participants gave 4 points regarding the willingness to re-enter the program (“will participate again”) and the recommendation to other nursing staff (“will recommend this program to other nursing staff”) on the 1–5 Likert scale.

## 4. Discussions

In this case series, we report the first attempt to apply an online mindfulness program to nursing staff in a KM clinic in Korea. Given the emphasis on the mental health of nursing staff [1,2,3,4] and the potential benefits of mind–body interventions, including mindfulness training [8], this topic is clinically relevant. Korea has a dual medical system, including WM and KM. Since the working environment and mental health-related factors of nursing staff working in these two institutions may differ [7], it is necessary to introduce a mental health improvement strategy tailored to their characteristics.

Given the context of COVID-19 and the working environment of the clinic, the program was conducted online from 8–10 p.m. (two hours) and consisted of five sessions for two months. Except for one participant who could not participate in the third session due to personal reasons, all others participated and showed a high participation rate in the program (14/15, 93.33%). Of the three female nursing assistants who participated in this two-month online mindfulness program, two had a decreased level of emotional labor while the other had an increased level of emotional labor. The increase in the emotional labor level was due to an increase in the subscale of employee-focused emotional labor. As suggested by Brotheridge et al., emotional labor can be largely divided into job-focused emotional labor (i.e., work demands regarding emotional expression) and employee-focused emotional labor (i.e., regulation of feelings and emotional expression) [20]. To date, no studies have reported a relationship between mindfulness interventions and job-focused or employee-focused emotional labor. It is also possible that other factors were involved in the increase in the participants’ emotional labor level. The increase in the level of employee-focused emotional labor in P2 was accompanied by an increase in the subscales of surface acting (from 3.33 to 3.67) and deep acting (from 3.33 to 4), among which the increase in deep acting was more pronounced. However, P3 showed no change in deep acting and P1 showed a decrease in deep acting (from 2 to 1.67), but this decrease was smaller than the decrease in surface acting (from 2.33 to 1). Deep acting, one aspect of emotional labor, requires effort to actually feel and express the required emotions, and some studies show that, unlike surface acting, deep acting is related to better mental health and high job satisfaction [21,22]. Thus, though P2 showed an increase in overall emotional labor level after the mindfulness program, given that deep acting was most affected, it is likely that the mindfulness program still had a positive effect on their mental health and job satisfaction.

Burnout levels were reduced in all three participants after this program. Although supported by limited-quality evidence, mind–body modalities, including mindfulness-based interventions, may reduce burnout in healthcare workers [23] and nursing staff [8]. In addition, this multi-purpose, trans-diagnostic intervention avoids the mental illness-related stigma of healthcare workers [24,25]. The program is also easy to apply at the group level, compared to some interventions (e.g., cognitive behavior therapy) that focus on mental health or specific mental symptoms. The participants expressed a relatively high level of satisfaction with this program in terms of recommendations for peers and willingness to participate again.

When implementing the program for the first time in this population, suggestions for improvement were also raised. For example, one participant reported difficulty concentrating during the program because it was implemented late. Although the program was introduced late in the evening to accommodate the schedules of the participants working at the clinic, this may not be the optimal time for mindfulness training for participants who are potentially exhausted from work. If this program was implemented during working hours, and a set amount of time for mindfulness was officially allocated by the institution, the employee’s participation in the program would be a part of the normal workday, and the difficulty in concentrating might be reduced. This program was introduced once every two weeks because it was judged that a session held late every week could potentially lower the interest and motivation of the participants and negatively affect their participation in the program. However, since the frequency of mindfulness sessions conducted in this program is relatively rare compared to a conventional mindfulness-based program, typically conducted once a week [26], the participants may not have experienced optimal benefits. Nevertheless, since bi-weekly mindfulness programs introduced online are currently being evaluated [27], the comparative benefits of the weekly or bi-weekly mindfulness program in this group may be investigated in the future. Shortened sessions of the mindfulness program in this case series should also be considered. Unlike the eight sessions in a typical MBSR, in this case series, participants were provided with a five-session mindfulness program. Although the minimal intensity and duration of mindfulness programs with meaningful benefits need to be further studied, it is worthwhile to examine the impact of abbreviated programs to increase their usefulness and efficiency [28]. Encouragingly, in a clinical study comparing an 8-week mindfulness training program with a 4-week abbreviated version in the non-clinical population, similar effect sizes were found in the groups on participants’ mindfulness, self-compassion, emotion, and resilience [29]. In addition, in a meta-analysis that analyzed the effect of brief mindfulness training lasting for less than 2 weeks, a modest effect size was found to reduce the negative emotions of participants [30]. The authors believe that it will be worthwhile to continue to experiment with abbreviated forms of mindfulness programs in order to successfully introduce mindfulness training in occupations exposed to busy work including healthcare workers.

The limitations of this study include the following. First, as this study is a retrospective case series, the number of participants is small and there was no control group in this study, so the possibility of generalization is highly limited. In other words, it is difficult to extend the results of this study beyond the meaning of the first case of applying online mindfulness training to nursing staff working in a KM clinic. Furthermore, the results of this study cannot be considered clinical evidence to prove the effectiveness of the mindfulness training program on nursing staff. Second, this case series did not assess the long-term impact of the program after its implementation. Moreover, although the last session of the program (i.e., Session 5) encouraged the participants to apply mindfulness and loving-kindness in their daily lives, participants’ adherence to daily mindfulness practice was not assessed after the program ended. In a randomized controlled clinical trial that compared the effects of 8- and 4-sessions of mindfulness training on a non-clinical population, no significant differences were found between groups of different mindfulness facets in six-month follow-ups [29]. However, self-kindness showed significantly better results in the eight-session group compared to the four-session group in six-month follow-ups [29]. Likewise, we cannot rule out the possibility that the program in this case series had a long-term effect on the participants, but this should be verified in a better-designed study in the future. Third, unlike the typical MBSR, the program includes compassion and loving-kindness sessions in equal proportions with those of mindfulness training. Mindfulness meditation and loving-kindness or compassion meditation are qualitatively distinct, and their expected effects may differ accordingly [13]. Moreover, among mindfulness-based interventions, there is a possibility that sitting meditation, body scan, and mindful yoga may have different effects on the participants [31]. However, since the program used in this case series is composed of heterogeneous mind–body exercises, the individual effects of each element cannot be estimated.

## 5. Conclusions

As this report is a case series, more studies are needed to fully evaluate the benefits of this program on emotional labor and burnout in nursing staff working in a KM clinic. Moreover, because the number of participants included was small and there was no control group, this study cannot be used to demonstrate the effectiveness of the online mindfulness program in nursing staff. However, based on this first attempt, our research team plans to expand the introduction of an online mindfulness program to improve the mental health of nursing staff working at KM clinics in Korea in the future. The potential benefits of emotional labor and burnout, feasibility, high satisfaction, and some challenges identified in this case series will be considered in future extensions and modifications of this program.

## Figures and Tables

**Figure 1 healthcare-11-00145-f001:**
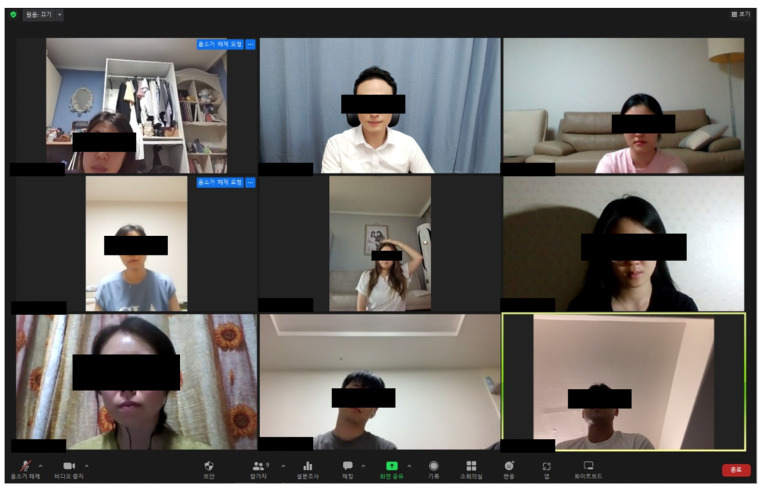
A screenshot of the online mindfulness program via Zoom meeting (Session 1).

**Figure 2 healthcare-11-00145-f002:**
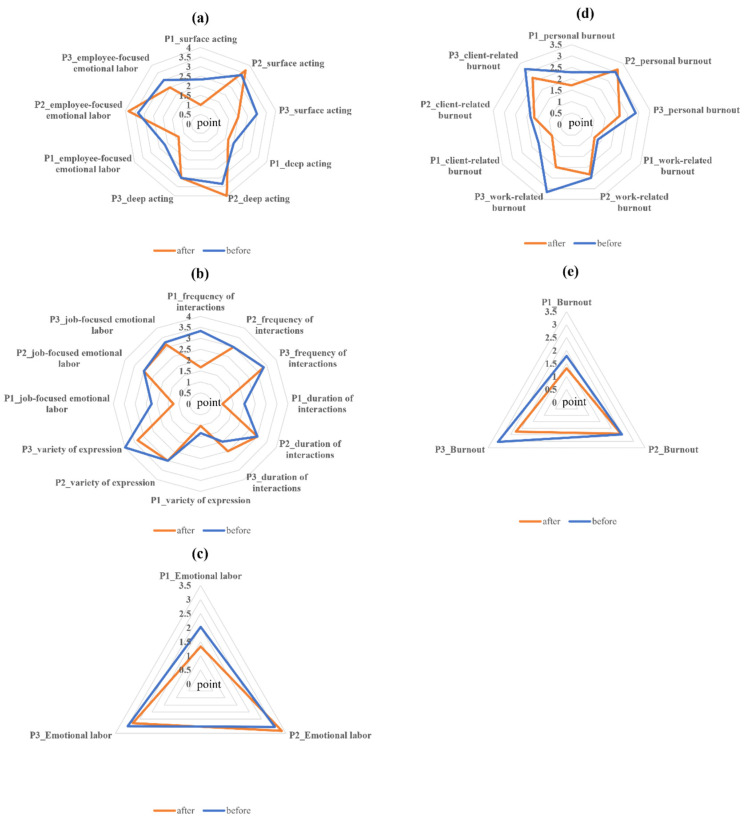
Pre-and-post evaluation of the participants. (**a**) Employee-focused emotional labor, (**b**) job-focused emotional labor, (**c**) total emotional labor, (**d**) subscales of burnout, and (**e**) total burnout in participants before and after the mindfulness program. P = participant.

**Table 1 healthcare-11-00145-t001:** Mindfulness program for the participants.

Session	Topic	Lessons
1 (Week 1)	Introduction to mindfulness	- Concept of mindfulness - Mindful eating and body scanning
2 (Week 3)	Advanced mindfulness	- The practice of mindfulness - Sitting meditation and mindfulness yoga
3 (Week 5)	Compassion	- Concept of compassion - Loving-kindness meditation
4 (Week 7)	Self-care	- Concept of compassion fatigue - Practice of self-care
5 (Week 9)	Work, love, and mindfulness	- Applying mindfulness in everyday life - Create individual action plans of mindfulness

## Data Availability

Original data are available upon request to the corresponding author.

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
