# Peer review of "The First Attempt to Apply an Online Mindfulness Program to Nursing Staff in a Traditional Korean Medicine Clinic in COVID-19 Era: A Case Series"

_healthcare, 2023, doi:10.3390/healthcare11010145_

Round 1

Reviewer 1 Report

The manuscript is focused on an interesting and current issue. As a case report, it uses small number of participants. An additional round of English editing would be beneficial; while grammatically correct, some sentences seem to be a bit awkward. Lack of comparison does not allow to attribute the change in outcomes to the interventions; it is possible that other non-participating nurses experienced same decline in negative effects over time as the intervention group.

Author Response

  • Response to Comments from Reviewer 1

Overall comment:

The manuscript is focused on an interesting and current issue. As a case report, it uses small number of participants. An additional round of English editing would be beneficial; while grammatically correct, some sentences seem to be a bit awkward. Lack of comparison does not allow to attribute the change in outcomes to the interventions; it is possible that other non-participating nurses experienced same decline in negative effects over time as the intervention group.

Response:            

Thank you very much for taking your valuable time to review this manuscript. We have no doubt that the reviewer’s comments will help to further improve the quality of this manuscript.

We fully agree with the reviewer's comment, and agree that the limitations of this study need to be further emphasized. Therefore, we added a paragraph in the Discussion section explaining the limitations of this study. The limitations include difficulty of generalization of this study results due to the study design and lack of evaluation of long-term effects.

“The limitations of this study include the followings. First, as this study is a retrospective case series, the number of the participants is small, and there was no control group in this study, the possibility of generalization is highly limited. In other words, it is difficult to extend the results of this study beyond the meaning of the first case of applying online mindfulness training to nursing staff working in a KM clinic. And the results of this study cannot be considered as clinical evidence to prove the effectiveness of the mindfulness training program on nursing staff. Second, this case series did not assess the long-term impact of the program after its implementation. Also, although the last session of the program (i.e., Session 5) encouraged the participants to apply mindfulness and loving-kindness in their daily lives, the participants' adherence to daily mindfulness practice was not assessed after the program ended. In a randomized controlled clinical trial that compared the effects of 8- and 4-sessions of mindfulness training on a non-clinical population, no significant differences were found between groups of different mindfulness facets in six-month follow-ups [29]. However, self-kindness showed significantly better results in the 8-session group compared to the 4-session group in six-month follow-ups [29]. Likewise, we cannot rule out the possibility that the program in this case series had a long-term effect on the participants, but this should be verified in a better designed study in the future. Third, unlike the typical MBSR, the program includes compassion and loving-kindness sessions in equal proportions with those of mindfulness training. Mindfulness meditation and loving-kindness or compassion meditation are qualitatively distinct, and their expected effects may differ accordingly [13]. Moreover, among mindfulness-based interventions, there is a possibility that sitting meditation, body scan, and mindful yoga may have different effects on the participants [31]. However, since the program used in this case series is composed of heterogeneous mind-body exercises, the individual effects of each element cannot be estimated.”

(Please refer to the red words on page 7)

Also, while reviewing the entire manuscript, we corrected some awkward sentences. All corrections are marked in red text. Nevertheless, if some sentences in this manuscript are still awkward for the reviewer, we are willing to make further corrections. Thank you.

Reviewer 2 Report

Please expand the Conclusions with more detailed explanations of limitations of the study (e.g., very low number of participants), the further ideas to be applied in real life and how could be expanded to other areas of life.

Please also check the attached file for minor language corrections.

Author Response

  • Response to Comments from Reviewer 2

Comment 1:

Please expand the Conclusions with more detailed explanations of limitations of the study (e.g., very low number of participants), the further ideas to be applied in real life and how could be expanded to other areas of life. 

Response:            

Thank you very much for taking your valuable time to review this manuscript. Based on the comment, we added a paragraph in the Discussion section explaining the limitations of this study, including difficulty of generalization of this study results due to the study design.

“The limitations of this study include the followings. First, as this study is a retrospective case series, the number of the participants is small, and there was no control group in this study, the possibility of generalization is highly limited. In other words, it is difficult to extend the results of this study beyond the meaning of the first case of applying online mindfulness training to nursing staff working in a KM clinic. And the results of this study cannot be considered as clinical evidence to prove the effectiveness of the mindfulness training program on nursing staff. Second, this case series did not assess the long-term impact of the program after its implementation. Also, although the last session of the program (i.e., Session 5) encouraged the participants to apply mindfulness and loving-kindness in their daily lives, the participants' adherence to daily mindfulness practice was not assessed after the program ended. In a randomized controlled clinical trial that compared the effects of 8- and 4-sessions of mindfulness training on a non-clinical population, no significant differences were found between groups of different mindfulness facets in six-month follow-ups [29]. However, self-kindness showed significantly better results in the 8-session group compared to the 4-session group in six-month follow-ups [29]. Likewise, we cannot rule out the possibility that the program in this case series had a long-term effect on the participants, but this should be verified in a better designed study in the future. Third, unlike the typical MBSR, the program includes compassion and loving-kindness sessions in equal proportions with those of mindfulness training. Mindfulness meditation and loving-kindness or compassion meditation are qualitatively distinct, and their expected effects may differ accordingly [13]. Moreover, among mindfulness-based interventions, there is a possibility that sitting meditation, body scan, and mindful yoga may have different effects on the participants [31]. However, since the program used in this case series is composed of heterogeneous mind-body exercises, the individual effects of each element cannot be estimated.”

(Please refer to the red words on page 7)

Also, the limitations of this study were added to the Conclusion section.

“As this report is a case series, more studies are needed to fully evaluate the benefits of this program on emotional labor and burnout in nursing staff working in a KM clinic. Moreover, because the number of participants included was small, and there was no control group, this study cannot be used to demonstrate the effectiveness of the online mindfulness program on the nursing staffs. However, based on this first attempt, our research team plans to expand the introduction of an online mindfulness program to improve the mental health of nursing staff working at KM clinics in Korea in the future. The potential benefits of emotional labor and burnout, feasibility, high satisfaction, and some challenges identified in this case series will be considered in future extensions and modifications of this program.”

(Please refer to the red words on pages 7-8)

Comment 2:

Please also check the attached file for minor language corrections.

Response:            

We especially appreciate this kind comment. We checked the attached file and corrected some expression according to the memo. All corrections are marked in red text.

Reviewer 3 Report

I appreciate this article and can see it can have relevance to medical practice in Korean contexts, especially with respect to the practice of traditional Korean medicine. 

While I would recommend publication, there are three areas of concern I think need to be addressed.

First, there is little theoretical conceptualization; that is, there is no theoretical framing of mindfulness that references literature on and about mindfulness--what it is (how it is defined or conceptualized), its origins, how it is practiced, what distinguishes it from other meditation practices, and so on. The paper jumps from the introduction right into a description of the program and the research methodology. The conceptual and practical distinctions of 'mindfulness yoga' and 'sitting meditation' are not clear. Are both part of mindfulness? As well, in the contemplative literature, both compassion meditation and loving-kindness meditation are distinguished from mindfulness meditation, per se. Having a well-developed theory (and literature review) section after the introduction would be helpful in informing readers about the details of mindfulness, compassion meditation and loving-kindness meditation. 

The lack of a theory section contributes, I think, to a methodological problem. It isn't clear whether the benefits came from the mindfulness practice or the loving-kindness meditation. Again, are you arguing that both are or should be part of mindfulness practice? If so, then you need to make that case in a theory section, preferably providing support from the literature for doing so. 

Second, I would suggest creating a section on the limitations of the study. All research studies have limitations, and we need to acknowledge these. As you have only a very small number of research subjects, you need to acknowledge that the findings are not generalizable, even though they can contribute to the growing body of knowledge. As well, we don't know the long-term impacts of your mindfulness program; it is not clear when the post-evaluation was performed, and this needs to be clarified. This is a significant problem in the research on the impacts of mindfulness and other contemplative practices--the lack of truly long-term longitudinal studies. Is it likely that 5 sessions of mindfulness practice are going to have a long-term impact? Do we know if the participants in this study did or will continue with their mindfulness practice and how regular will that practice be? That would be worth investigating in future research. A Kabat-Zinn study from 1982 is cited to support once-a-week practice, but what does more current literature indicate about frequency of practice and how that relates to effectiveness? 

Finally, I think the article needs more about the ethical parameters of the research (with appropriate references to the literature on ethical practices in human subject research). Was the consent given a fully informed consent? Were the participants informed they could withdraw their consent? What efforts were made to maintain confidentiality? How were the data kept secure? 

I wish you well in pursuing this worthy line of research! 

Author Response

  • Response to Comments from Reviewer 3

Comment 1:

I appreciate this article and can see it can have relevance to medical practice in Korean contexts, especially with respect to the practice of traditional Korean medicine.

While I would recommend publication, there are three areas of concern I think need to be addressed.

Response:            

Thank you very much for taking your valuable time to review this manuscript. We have no doubt that the reviewer’s comments will help to further improve the quality of this manuscript.

Comment 2:

First, there is little theoretical conceptualization; that is, there is no theoretical framing of mindfulness that references literature on and about mindfulness--what it is (how it is defined or conceptualized), its origins, how it is practiced, what distinguishes it from other meditation practices, and so on. The paper jumps from the introduction right into a description of the program and the research methodology. The conceptual and practical distinctions of 'mindfulness yoga' and 'sitting meditation' are not clear. Are both part of mindfulness? As well, in the contemplative literature, both compassion meditation and loving-kindness meditation are distinguished from mindfulness meditation, per se. Having a well-developed theory (and literature review) section after the introduction would be helpful in informing readers about the details of mindfulness, compassion meditation and loving-kindness meditation.

The lack of a theory section contributes, I think, to a methodological problem. It isn't clear whether the benefits came from the mindfulness practice or the loving-kindness meditation. Again, are you arguing that both are or should be part of mindfulness practice? If so, then you need to make that case in a theory section, preferably providing support from the literature for doing so.

Response:            

Thank you for the comment. We fully agree with the reviewer's comment about the lack of theoretical conceptualization in the original version of the manuscript. In this revised manuscript, the following corrections were made according to the comments.

  1. The theoretical conceptualization was strengthened in the Introduction part. In particular, explanations on mindfulness, loving-kindness meditation and compassion meditation have been highlighted.

“Mindfulness is defined as “the awareness that emerges through paying attention on purpose, in the present moment, and nonjudgmentally to the unfolding of experience moment by moment” [9], by Kabat-Zin, who developed mindfulness-based stress reduction (MBSR) program. In some systematic reviews, MBSR was reported to be effective in improving mindfulness and self-compassion, and reducing anxiety, depression, and stress in healthcare professionals [10] as well as employees [11]. Although closely related to mindfulness-based meditation and often included in modified MBSR program [12], there are loving-kindness meditation and compassion meditation as distinct meditation practices [13]. These meditations are exercises to strengthen the emotional state of kindness and compassion, and their clinical benefits are supported by neuroendocrine studies, neuroimaging studies, and preliminary clinical studies [13]. Given the work specificity of healthcare professionals, it has been suggested that both mindfulness and loving-kindness meditation can help them reduce compassion fatigue and burnout, increase their well-being, and strengthen their relationships with clients [14,15].”

(Please refer to the red words on page 2)

  1. We have added explanations and distinctions between mindfulness yoga and sitting meditation.

“In Session 2, participants practiced sitting meditation (static practice) and mindfulness yoga (dynamic practice). Mindfulness yoga is a kind of meditative movement that emphasizes awareness. Sitting meditation is a typical meditation practice that raises awareness of the five senses, emotions, and thoughts non-judgmentally. Both sitting meditation and yoga exercise are considered the formal practices of MBSR.”

(Please refer to the red words on pages 3-4)

  1. In this revised manuscript, we described the inclusion of both mindfulness and non-mindfulness practice (i.e., loving-kindness and compassion meditation) within the program, and added limitations that arise accordingly.

“The program was created by certified professionals with PhDs in psychology who have been running these programs for working professionals for many years. In general, this program originates from MBSR, but there are some differences in its content and implementation. Specifically, the program was modified based on the analysis of mental health factors in the Korean nursing staffs previously investigated by our team [5,7,17] and the practitioner’s experience. In the program, compassion and loving-kindness were emphasized, and 3 out of 5 sessions (60%) included the topic. Meanwhile, among the formal practices of MBSR, walking meditation was excluded from this program as it was considered inappropriate due to the characteristics of the online non-face-to-face program. Each session was conducted bi-weekly, for a total of five sessions, and it was held for two hours from 8–10 p.m., accommodating the nurses’ working schedule at the local KM clinic. It is less frequent and shorter in duration than the MBSR, which typically consists of eight 2.5-hour classes per week and one 8-hour all-day class.”

(Please refer to the red words on pages 2-3)

“The limitations of this study include the followings. Third, unlike the typical MBSR, the program includes compassion and loving-kindness sessions in equal proportions with those of mindfulness training. Mindfulness meditation and loving-kindness or compassion meditation are qualitatively distinct, and their expected effects may differ accordingly [13]. Moreover, among mindfulness-based interventions, there is a possibility that sitting meditation, body scan, and mindful yoga may have different effects on the participants [31]. However, since the program used in this case series is composed of heterogeneous mind-body exercises, the individual effects of each element cannot be estimated.”

(Please refer to the red words on page 7)

Comment 3:

Second, I would suggest creating a section on the limitations of the study. All research studies have limitations, and we need to acknowledge these. As you have only a very small number of research subjects, you need to acknowledge that the findings are not generalizable, even though they can contribute to the growing body of knowledge. As well, we don't know the long-term impacts of your mindfulness program; it is not clear when the post-evaluation was performed, and this needs to be clarified. This is a significant problem in the research on the impacts of mindfulness and other contemplative practices--the lack of truly long-term longitudinal studies. Is it likely that 5 sessions of mindfulness practice are going to have a long-term impact? Do we know if the participants in this study did or will continue with their mindfulness practice and how regular will that practice be? That would be worth investigating in future research. A Kabat-Zinn study from 1982 is cited to support once-a-week practice, but what does more current literature indicate about frequency of practice and how that relates to effectiveness?

Response:            

We especially appreciate this valuable comment. Following the reviewer's comments, we added a paragraph in the Discussion section explaining the limitations of this study. The limitations include difficulty of generalization of this study results due to the study design, lack of evaluation of long-term effects, and possibility of long-term impact by the 5-session mindfulness practice.

“The limitations of this study include the followings. First, as this study is a retrospective case series, the number of the participants is small, and there was no control group in this study, the possibility of generalization is highly limited. In other words, it is difficult to extend the results of this study beyond the meaning of the first case of applying online mindfulness training to nursing staff working in a KM clinic. And the results of this study cannot be considered as clinical evidence to prove the effectiveness of the mindfulness training program on nursing staff. Second, this case series did not assess the long-term impact of the program after its implementation. Also, although the last session of the program (i.e., Session 5) encouraged the participants to apply mindfulness and loving-kindness in their daily lives, the participants' adherence to daily mindfulness practice was not assessed after the program ended. In a randomized controlled clinical trial that compared the effects of 8- and 4-sessions of mindfulness training on a non-clinical population, no significant differences were found between groups of different mindfulness facets in six-month follow-ups [29]. However, self-kindness showed significantly better results in the 8-session group compared to the 4-session group in six-month follow-ups [29]. Likewise, we cannot rule out the possibility that the program in this case series had a long-term effect on the participants, but this should be verified in a better designed study in the future. Third, unlike the typical MBSR, the program includes compassion and loving-kindness sessions in equal proportions with those of mindfulness training. Mindfulness meditation and loving-kindness or compassion meditation are qualitatively distinct, and their expected effects may differ accordingly [13]. Moreover, among mindfulness-based interventions, there is a possibility that sitting meditation, body scan, and mindful yoga may have different effects on the participants [31]. However, since the program used in this case series is composed of heterogeneous mind-body exercises, the individual effects of each element cannot be estimated.”

(Please refer to the red words on page 7)

Also, in the Discussion section, reports of the current literature according to the frequency, duration, and intensity of mindfulness training have been added.

“Nevertheless, since bi-weekly mindfulness programs introduced online are currently being evaluated [27], the comparative benefits of the weekly or bi-weekly mindfulness program in this group may be investigated in the future. Shortened sessions of the mindfulness program in this case series should also be considered. Unlike the 8 sessions in a typical MBSR, in this case series, participants were provided with a 5-session mindfulness program. Although the minimal intensity and duration of mindfulness programs with meaningful benefits need to be further studied, it is worthwhile to examine the impact of abbreviated programs to increase their usefulness and efficiency [28]. Encouragingly, in a clinical study comparing 8-week mindfulness training program with 4-week abbreviated version in the non-clinical population, similar effect sizes were found in the groups on participants' mindfulness, self-compassion, emotion, and resilience [29]. In addition, in a meta-analysis that analyzed the effect of brief mindfulness training lasting for less than 2 weeks, a modest effect size was found to reduce negative emotions of participants [30]. The authors believe that it will be worthwhile to continue to experiment with abbreviated forms of mindfulness program in order to successfully introduce mindfulness training in occupations exposed to busy work including healthcare workers.”

(Please refer to the red words on page 7)

In this revised manuscript, we clearly described the timing of pre- and post-evaluation.

“Evaluation results were collected twice (1 day before Session 1 and 1 day after Session 5) using Google Forms, and extracted to a Microsoft Excel form (Microsoft, USA). The participants' data were visualized using Microsoft Excel software.”

(Please refer to the red words on page 4)

Comment 4:

Finally, I think the article needs more about the ethical parameters of the research (with appropriate references to the literature on ethical practices in human subject research). Was the consent given a fully informed consent? Were the participants informed they could withdraw their consent? What efforts were made to maintain confidentiality? How were the data kept secure?

I wish you well in pursuing this worthy line of research!

Response:            

Thank you for the comment. We have added some descriptions about the ethical reporting to this case series as follows.

“This case series complied with the 2013 CARE checklist [16]. All subjects voluntarily participated in the online mindfulness program with an instructor present. The participants were informed that the results of the pre-post evaluation would be used for research purposes, and participation in this evaluation was not mandatory for participation in the program. The participants were informed that they could withdraw from the program or withdraw from providing data for the study at any time they wished. The participants consented to the use of personal information and participated in the evaluation, and the data was collected online and anonymized. Fully informed consent from participants was obtained twice, through an online survey platform, Google Forms (Google, USA), before the program and in writing after the program.”

(Please refer to the red words on page 2)

The consent form received from the participants has already been submitted at the request of the editor of this journal. However, I have attached it as non-published material so that the reviewer can also check it.

Round 2

Reviewer 3 Report

A much improved version! I especially appreciate the section on the study's limitations. I am now willing to 'sign off' on this version. I wish you all the best with your future research efforts--you are doing great work!